# Exposure to Ionizing Radiation Triggers Prolonged Changes in Circular RNA Abundance in the Embryonic Mouse Brain and Primary Neurons

**DOI:** 10.3390/cells8080778

**Published:** 2019-07-26

**Authors:** André Claude Mbouombouo Mfossa, Helene Thekkekara Puthenparampil, Auchi Inalegwu, Amelie Coolkens, Sarah Baatout, Mohammed A. Benotmane, Danny Huylebroeck, Roel Quintens

**Affiliations:** 1Radiobiology Unit, Belgian Nuclear Research Centre, SCK·CEN, 2400 Mol, Belgium; 2Department of Development and Regeneration, KU Leuven, 3000 Leuven, Belgium; 3Department of Cell Biology, Erasmus University Medical Center, 3015 CN Rotterdam, The Netherlands

**Keywords:** circular RNA, ionizing radiation, P53 targets, embryonic brain, neuronal maturation

## Abstract

The exposure of mouse embryos in utero and primary cortical neurons to ionizing radiation results in the P53-dependent activation of a subset of genes that is highly induced during brain development and neuronal maturation, a feature that these genes reportedly share with circular RNAs (circRNAs). Interestingly, some of these genes are predicted to express circular transcripts. In this study, we validated the abundance of the circular transcript variants of four P53 target genes (*Pvt1*, *Ano3*, *Sec14l5,* and *Rnf169*). These circular variants were overall more stable than their linear counterparts. They were furthermore highly enriched in the brain and their transcript levels continuously increase during subsequent developmental stages (from embryonic day 12 until adulthood), while no further increase could be observed for linear mRNAs beyond post-natal day 30. Finally, whereas radiation-induced expression of P53 target mRNAs peaks early after exposure, several of the circRNAs showed prolonged induction in irradiated embryonic mouse brain, primary mouse cortical neurons, and mouse blood. Together, our results indicate that the circRNAs from these P53 target genes are induced in response to radiation and they corroborate the findings that circRNAs may represent biomarkers of brain age. We also propose that they may be superior to mRNA as long-term biomarkers for radiation exposure.

## 1. Introduction

The exposure of cells and organisms to ionizing radiation is a well-known inducer of DNA-damage, of which DNA double-strand breaks (DSBs) are the most lethal form [1]. The cellular response to DSBs is a tightly regulated molecular mechanism that involves cell cycle arrest to allow for cells to repair the DNA, after which different outcomes are possible including cell cycle reentry, apoptosis and senescence [2]. Central in this so-called DNA-damage response (DDR) is P53, a tumor suppressor transcription factor that is activated via several post-translational modifications and regulates the expression of genes that are involved in each of the aforementioned processes. Typical for the P53-mediated transcriptional response to DNA-damage is the transient nature of gene expression changes [3], which often coincides with the levels of residual DSBs. The gene expression levels of P53 target genes quickly return to their respective basal levels once DNA-repair has been completed.

In a previous study, we have observed that exposure to radiation of mouse embryos and immature primary mouse cortical neurons leads to the rapid activation of the P53-mediated transcriptional response [4]. This results in the induction of a number of genes that are involved in some of the canonical P53-dependent pathways (cell cycle arrest, DNA-repair, apoptosis), as well as genes that were found not to be related to the classical P53 response. A unifying feature of this radiation-induced gene signature was that they are, in general, upregulated during brain development in vivo and during maturation of primary neurons in vitro. Although this is atypical for genes that are involved in those pathways [5], such an expression profile is very similar to that of the class of circular RNAs (circRNAs) that are enriched in the brain and dynamically upregulated during brain development and neuron differentiation and maturation [6,7].

CircRNAs are covalently closed RNAs, which lack a 5’-cap and a 3’-poly-A tail in comparison to linear mRNAs. They are mostly transcribed by RNA Polymerase-II (RNAPolII) from both protein-coding and non-coding genes and generated by the canonical splicing machinery from the pre-mRNA through “back-splicing” [8,9]. CircRNAs are widely expressed and, similar to linear splice variants, their expression profiles can vary according to the cell/tissue-type and developmental timing and age [10,11,12]. Many circRNAs are conserved between worms, flies and mammals [6,12,13], which suggests that they might have evolutionarily conserved functions. Although thousands of circRNA molecules have been identified, few have yet been functionally characterized [14]. Their most important functions are that they act as sponges for microRNAs [15,16] or RNA-binding proteins [17], and as regulators of expression [18] or alternative splicing [8] of their host gene. Finally, although they are mostly considered as non-coding RNAs, some circRNAs have been shown to have coding potential via 5’-cap independent translation of an open reading frame if an internal ribosome entry sequence is also present [19,20]. Interestingly, a Gene Ontology enrichment analysis of circRNA producing host genes that are expressed in mouse preimplantation embryos showed strong enrichment for terms that are related to chromatin organization, cell division and response to DNA-damage stimulus, which suggests the potential roles and fundamental evolutionary importance of these circRNAs [21].

We investigated whether some of these P53 target genes transcribe circRNAs because of the abovementioned similarities between the expression profiles of our radiation-responsive gene signature and circRNAs. Via RT-qPCR using divergent primers, the depletion of linear RNA using the exonuclease RNaseR, and by comparing cDNA synthesis using oligo-dT and random hexamer primers, we validated the transcript levels of various circRNAs that were derived from *Pvt1*, *Ano3*, *Sec14l5,* and *Rnf169*. We show that these circRNAs are, in general, more stable, highly enriched in the brain, and strongly upregulated until adulthood, while their peak steady-state levels are prolonged after exposure to ionizing radiation in embryonic brains and primary cortical neurons. Hence, the expression of circRNAs from P53 target genes is independently regulated from their linear counterparts; therefore, circRNAs with increased stability may represent a new class of transcriptional biomarkers of radiation exposure.

## 2. Materials and Methods

### 2.1. Animals

The animals were housed at the animal facility of SCK·CEN in accordance with the Ethical Committee Animal Studies of Medanex Clinic (EC_MxCl_2014_036). All of the animal experiments were carried out in compliance with the Belgian laboratory animal legislation and the European Communities Council Directive of 22 September 2010 (2010/63/EU). Wild-type C57BL/6J mice were purchased at The Jackson Laboratory and housed in the specific pathogen-free animal facility at SCK·CEN. The animals were held under controlled conditions and a standard 12:00/12:00 light/dark cycle was maintained. Food and water were available ad libitum. The mice were coupled only during a short period between 08:00 and 10:00 in the morning and the day of fertilization is referred to as embryonic day 0 (E0) to ensure synchronous timing of embryonic development. For in vivo experiments, the brains of mice at either E11, E12, and E18 or post-natal male mice at the ages of 10 days (P10), 30 days (P30), and six months were used. To compare the expression in different tissues, adult male mice (>6 months of age) were used. Adult female mice (>6 months of age) were used for experiments on blood samples.

### 2.2. Cell Culture and Transfection

15 days C57BL/6J pregnant females were sacrificed by cervical dislocation for primary cortical neurons culture. The fetuses were rapidly placed in Hanks’ Balanced Salt solution (HBSS) medium containing 10 mM HEPES before brain cortex dissection. The isolated cortices were pooled, single primary cortical neuronal cells were obtained by incubation with 0.1% Trypsin (Invitrogen) for 20 min. at 37 °C. At the end of the incubation, the enzymatic solution was gently removed and replaced by the plating medium containing a minimum essential medium, 1 mM sodium pyruvate, 0.6% glucose, and 10% fetal bovine serum. The cells were plated at a density of 3 × 10^5^ cells/wells in six-well poly-d-lysine-coated plates (Corning BioCoat) and placed at 37 °C for 90 min. in a humidified incubator containing 5% CO2. At the end of the incubation, the plating medium was completely replaced with the serum-free growth medium, consisting of Neurobasal medium with 2% B27 supplement, 2.5 mM glutamine. Every three days, half of the neurobasal medium was replaced with the same fresh pre-warmed medium.

Neuro2a mouse neuroblastoma cells were maintained in Dulbecco’s modified Eagle’s medium (Gibco, Aalst, Belgium) that was supplemented with 10% fetal bovine serum (Gibco) and 1× non-essential amino acids (Gibco), in a humidified incubator with 5% CO_2_ at 37 °C. The cells were routinely passaged at approximately 10^6^ cells/25 cm^2^ culture flask, and sub-cultured at a 1:8 ratio in fresh culture medium every 2 to 3 days.

Gene silencing was performed by transfecting Neuro2a cells with 10 nM of short interfering RNA against Qki (Dharmacon, CO, USA) using Lipofectamine 3000 (Thermoscientific, Aalst, Belgium). The cells were transfected on the day of subculturing in 500 µL of culture medium per well of a 24-well cell culture plate, according to the manufacturer’s instructions. RNA extraction was performed at 24 h post-transfection.

### 2.3. X-Irradiation

The irradiations were performed at the Laboratory for Nuclear Calibrations (LNK) of the Belgian Nuclear Research Centre (SCK·CEN) while using H-250 beam quality, as defined in ISO 4037 standard. This beam quality is created on the Xstrahl 320 kV tube of LNK using the following beam parameters: air kerma (K_air_) rate: 0.13 Gy/min; inherent filtration: 3 mm of Be; additional filtration: 3.8 mm Al + 1.4 mm Cu + Dose Area Product monitor ionizing chamber; tube voltage: 250 kV; tube current: 12 mA; distance focal point—sample center: 100 cm; beam orientation: vertical; and, beam diameter defined as Full Width at Half Maximum: 31 cm. The secondary standard K_air_ measurements are traceable to international standards, in accordance with ISO 17025 accreditation of LNK. The samples are always smaller than the beam diameter. For the in vivo studies, conscious pregnant mice at gestational day 11 (E11) were given a single dose of whole body irradiation (K_air_ = 1 Gy) while being individually placed in an octant of a custom-made, plexiglass irradiation pie. For in vitro experiments, primary neuronal cultures from brain cortices of E15 mice were irradiated (1 Gy) 24 h after being in culture (DIV1), while using the same setup as for the in vivo experiments. Control (0 Gy) mice or primary neuronal cultures were also transported to the irradiation facility, but they were not placed within the radiation field (sham-irradiation).

### 2.4. RNA Extraction, cDNA Synthesis, and RT-qPCR

Total RNA was purified from embryonic, post-natal and adult tissue samples, embryonic brain cortices at 2 h, 6 h, 12 h post-irradiation, as well as primary cortical neurons at 6 h, 24 h, and 48 h post-irradiation while using the AllPrep DNA/RNA Mini Kit (Qiagen, Venlo, The Netherlands), according to provided instructions. Total RNA from mouse blood was extracted while using the Direct-zol RNA Miniprep kit (Zymo Research, Irvine, CA, USA). The concentration of the purified RNA was determined using a spectrophotometer (Trinean Xpose™, Proteigene, Saint-Marcel, France). The GoScript™ Reverse Transcription System (Promega, Leiden, The Netherlands) was used according to provided instructions to prepare cDNA from the purified RNA. For RT-qPCR, each individual sample of biological replicates was run in technical duplicates while using the MESA GREEN^®^ RT-qPCR kit (Eurogentec, Seraing, Belgium) on an Applied Biosystems^®^ 7500 Real-Time PCR instrument, following the manufacturer’s instructions. The specificity of the primers was confirmed using a melting curve. Details to all of the primers used are listed in Appendix A. Gapdh and/or Polr2a served as internal reference genes to correct for differences in the RNA input for cDNA synthesis. The reaction efficiencies were used for relative quantification using the method, as described by Pfaffl [22].

### 2.5. Ribonuclease-R (RNaseR) Treatment

RNA purified from mouse brain at P10 was treated with RNaseR (Epicentre RNR07250) at concentrations of 1, 3, and 10 U of RNaseR/μg of total RNA. A non-treated sample served as control. The reaction mix that included RNA (2 μg of total RNA), 10x RNaseR Buffer, and RNaseR was incubated at 37 °C for 15 min. and subsequently processed for purification using the AllPrep DNA/RNA Mini Kit (Qiagen), according to provided instructions.

### 2.6. Actinomycin-D Treatment

Primary cortical neuronal cultures after seven days in vitro (DIV7) were treated with Actinomycin-D (solubilized in 1% DMSO) at a concentration of 1 μg/mL for a maximum of 24 h. RNA-extractions were performed 1 h, 2 h, 4 h, 8 h, and 24 h after the addition of Actinomycin-D to the cell culture medium. Cultures that were treated with only DMSO (1%) served as the timepoint zero control.

### 2.7. Statistical Analysis

All of the results were analyzed while using Graphpad Prism 7.02. The statistical tests used for individual experiments are indicated in figure legends.

## 3. Results

Previously, we have mapped a p53-dependent radiation-induced gene signature in the embryonic mouse brain [4]. Now, we aimed at investigating the expression patterns of possible circRNA variants for a number of these genes. We selected *Pvt1*, a long non-coding (lnc) RNA of which the human ortholog is implicated in cancer progression [23]; *Ano3*, a member of the *Tmem16* family of membrane proteins and a possible potassium channel [24]; *Sec14l5*, an uncharacterized gene; and, *Rnf169*, which plays a role in the DDR via interaction with DYRK1A [25].

### 3.1. Validation of Predicted Non-Coding Circular Transcripts of P53-Regulated Genes

According to the circbase database for circular RNAs (www.circbase.org), 5, 5, 1, and 4 circular transcripts are predicted to exist for *Pvt1*, *Ano3*, *Sec14l5*, and *Rnf169*, respectively (Figure 1). Three approaches were employed in order to verify their expression. First, while using outward-directed or divergent primers spanning the predicted circular backsplice sites (Appendix A), RT-qPCR was performed on the total RNA samples that were extracted from mouse brain tissues at different developmental stages (embryonic day (E) 11, E18 and adult age). The results showed that for both *Pvt1* and *Rnf169*, clear amplification was observed for three predicted circular transcripts, while for *Ano3,* four out of five predicted circular variants generated a product. For *Sec14l5*, one circular transcript was predicted, for which a distinct amplicon was generated.

Secondly, based on exonuclease resistance of circRNAs (due to the absence of 5’ and 3’ ends [26]), RT-qPCR analyses were performed on total RNA samples that were extracted from mouse brains that had been digested with different concentrations of RNaseR. This resulted in a dose-dependent degradation of the linear mRNAs, as well as the control Gapdh mRNA. The maximal depletion of mRNA was found at a dose of 3 U RNaseR/µg total RNA. In contrast, the circular transcripts indeed showed resistance to RNaseR treatment, which led to an apparent enrichment of most of the circRNAs with higher RNaseR concentrations (Figure 2a–d).

Lastly, RT-qPCR was performed on RNA samples that had been reverse transcribed using either oligo-dT primers or random hexamer primers to distinguish between the linear mRNAs containing a poly-A tail and circRNAs. This showed that the efficient amplification of both linear and circular transcripts could be achieved when using random primers, while almost no PCR product was produced for the circular candidates when using th oligo-dT primers (Figure 2e–h). Importantly,, the amplification of the linear transcripts of *Ano3* (Figure 2f) and Sec14l5 (Figure 2g) was less efficient when using the random primers in comparison with the oligo-dT primers, unlike for *Pvt1* (Figure 2e) and *Rnf169* (Figure 2h). This can be explained by the fact that we used RT-qPCR primers targeting the 3’ part of the transcript for Lin_Pvt1 and Lin_Rnf169, while the primers were directed to the 5’ part for Lin_Ano3 and Lin_Sec14l5. Thus, the reverse transcription of the 3’ part of transcripts is more efficient using oligo-dT primers, while the 5’ part can be more efficiently reverse transcribed while using random primers.

### 3.2. Circular Transcripts are More Stable Compared to Linear Cognates in Primary Cortical Neurons

CircRNAs are more stable than most linear RNAs in general due to their inaccessibility to exonucleases [27]. mRNA stability was assayed by treating primary cortical neurons with Actinomycin-D, an antibiotic that intercalates to DNA thereby blocking RNAPolII-mediated transcription, to assess whether this was also the case for our genes of interest. The stability of control mRNAs was as expected; Gapdh mRNA has a long half-life (>24 h), while that of less stable transcripts such as *Myc* and *Tbp* was <2 h (Figure 3a) [27]. Pvt1 mRNA was also relatively stable although two of its circular variants (circ-0000604 and circ-0005724) showed increased stability (Figure 3b). The linear transcripts of *Ano3* and *Rnf169* were unstable (t1/2, <4 h, and <2 h, respectively), unlike their very stable circular transcripts (Figure 3c,d). The increased abundance of circ-0009174 in cells treated with Actinomycin-D (Figure 3c) suggests that the biogenesis of this transcript occurred, at least in part, post-transcriptionally, and therefore independent of the transcription machinery. Unfortunately, the expression of the circular variant of *Sec14l5* (circ-0006330) could not be detected in primary cortical neurons. Altogether, these results confirm an overall increased stability of circular transcripts when compared to their linear counterparts.

### 3.3. Tissue-Specific Presence of Circular Transcripts is Independent of Their Host mRNAs and is Enriched in the Brain

Although they originate from the same gene, it is known that the expression pattern of circRNAs can differ from their linear RNA counterparts [28]. We performed RT-qPCR while using RNA sampled from eight different tissues from adult (brain, eye, heart, kidney, liver, lung, spleen, and white adipose tissue, respectively). We found for *Pvt1* that its circular transcripts were highly enriched in the brain, whereas the linear transcript was preferentially expressed in the liver (Figure 4a–d). In the case of *Ano3*, the circular and linear transcripts were both enriched in the brain when compared to the other tissues (Figure 4e–i). The circRNA variant of *Sec14l5* was enriched in both brain and heart, whereas the linear mRNA was brain-specific (Figure 4j–k). Finally, the circular transcripts of *Rnf169* were also mostly enriched in the brain, although the linear mRNA was enriched in the liver (Figure 4l–o). In all, these results confirm that the expression of circRNAs is enriched in the brain [6,7,29] and that their expression patterns deviate from that of their cognate mRNAs, which indicates that they are independently regulated.

### 3.4. Steady-State Levels of circRNA are Enriched During Pre- and Post-Natal Brain Development and Neuronal Maturation

Besides being enriched in the brain, circRNAs are also upregulated during brain development and neuronal maturation [6,7], As these are features that are shared with our previously identified radiation response gene signature [4], we documented the linear and circular transcripts of our genes of interest during these two processes. Therefore, we performed RT-qPCR while using RNA extracted from brains at pre- and post-natal developmental stages (from E12 until adulthood), as well as in primary cortical neurons at different days in vitro (DIV1 until DIV14). This showed that the expression of *Pvt1* transcripts increased during embryonic brain development. However, while the level of circular variants kept increasing beyond P10, the expression of its mRNA was strongly reduced between P10 and P30 (Figure 5a). For *Ano3* and *Sec14l5*, the circRNAs levels dramatically increased between P10 and P30, while their mRNA expression remained stable between P30 and adult age, and that of the circRNAs further increased (Figure 5b,c). The *Rnf169* mRNA levels remained stable overall during brain development. In contrast, the abundance of its circRNAs increased until adult age, although a temporary reduction in transcript levels occurred between E18 and P10 for circ-0001605 and circ-0013703 (Figure 5d). In all, these results show that, while mRNA expression of these P53 target genes remains unchanged beyond P30, that of circular transcripts keeps increasing during the aging of the brain.

During in vitro neuronal maturation, *Pvt1* mRNA expression increased 5.2-fold between DIV1 and DIV7, subsequently reducing until DIV14 (Figure 6a). In contrast, the circ-0000604 and circ-0005721 increased by 2.4-fold and 3.9-fold between DIV1 and DIV7, after which they remained stable until DIV14. circ-0005724 expression peaked at DIV10 (2.1-fold) (Figure 6a). *Ano3* mRNA increased between DIV1 and DIV7 (3.3-fold) while circ-0009172, circ-0009174, and circ-0009175 primarily increased between DIV10 and DIV14 (Figure 6b). The expression of circ-0009176 did not substantially change during neuronal maturation. *Rnf169* mRNA, as well as circ-0013703 and circ-0013704, reduced during neuronal maturation, whereas circ-0001605 did not change.

### 3.5. Prolonged Upregulation of circRNA Transcript Levels in the Embryonic Brain and Primary Cortical Neurons

The selected genes had previously been identified on the basis of their induction by radiation exposure in the brain of mouse embryos and in primary mouse cortical neurons [4]. Therefore, we investigated whether the circular variants of these genes were also induced by radiation. We also analyzed RNA expression after extended time points following irradiation because circRNAs are more stable and radiation-induced p53-dependent transcriptional changes are generally short-lived. This showed that in vivo the mRNA expression of *Ano3*, *Sec14l5,* and *Rnf169* peaked at 2 h after irradiation, while *Pvt1* mRNA expression was the highest after 6 h (Figure 7a–d). Regarding the circRNAs, those of *Pvt1*, *Ano3,* and *Rnf169* (except for circ-0013704) showed the highest levels at 12 h after irradiation (Figure 7a,b,d). The circular isoform of *Sec14l5* showed a 13.8-fold induction already after 2 h, which remained unchanged until after 12 h (Figure 7c). Similarly, in primary cortical neurons mRNAs of *Pvt1*, *Ano3*, *Rnf169* (Figure 8a–c), and *Sec14l5* (data not shown) were induced at 6 h after irradiation, and then decreased after 24 and 48 h. Among the circRNAs, the large majority (8 out of 11) were significantly increased after 24 h and four of them remained upregulated after 48 h (Figure 8a–c). Therefore, these results show that the majority of the investigated circular transcripts of P53 targets are also induced after radiation exposure, albeit with different kinetics as their linear cognates, which suggested independently acting regulations.

### 3.6. Long-Term Induction of Pvt1 circRNA Abundance in Mouse Blood after Whole Body Irradiation

The prolonged induction of circRNAs after irradiation suggested that they could have improved potential as retrospective biomarkers of radiation exposure when compared to mRNAs, on which the effect of radiation is in general short lived [3]. We analyzed the transcript abundance in blood samples from mice at 10 days after whole body irradiation because brain tissue is not readily accessible for the purpose of biodosimetry. This showed that, while the expression of Pvt1 mRNA was unchanged, the circular Pvt1 transcripts were significantly more abundant (between 2 and 2.5-fold) in blood from irradiated mice (Figure 9a). In contrast, both linear and circular Sec14l5 transcripts were reduced after irradiation (Figure 9b), while for *Rnf169* only the linear form was reduced and the circRNAs remained unchanged (Figure 9c). Ano3 mRNA and circRNA could not be detected in mouse blood. Thus, long-term changes in transcript abundance after whole body irradiation seem to affect both linear and circular RNAs, although the increases were only observed among circRNAs that were transcribed from *Pvt1*.

### 3.7. Expression of Pvt1 circRNAs in Neuro-2a Cells Is Independent of the circRNA Biogenesis Regulator Quaking

One potential mechanism of circRNA biogenesis is via the binding of RNA-binding proteins to intron sequences spanning circRNA-forming exons. One such RNA-binding protein is Quaking (QKI), which has been shown to regulate the formation of hundreds of circRNAs during epithelial-to-mesenchymal transition (EMT) [30]. This process is related to cellular differentiation and it is important for embryonic development and cancer cell invasiveness. One of the pre-mRNAs to which QKI was found to bind to promote circRNA biogenesis was PVT1 [30]. The long non-coding RNA PVT1 has repeatedly been shown to promote EMT in various human cancer cell lines [31,32]. Therefore, we investigated whether this Qki-Pvt1 axis is also functional in mouse cells by knocking down the three major Qki isoforms (Qki-5, Qki-6, Qki-7) using siRNA in Neuro-2a neuroblastoma cells. Although all Qki isoforms could be effectively knocked down (Figure 10a), no effect on *Pvt1* circRNA expression was observed (Figure 10b), which indicated that, at least in Neuro-2a cells, Qki does not regulate *Pvt1* circRNA biogenesis.

## 4. Discussion

In this study, we have validated the existence of predicted circRNAs that were transcribed from four selected P53 target genes. Although our study focused on circular transcripts from only four genes, we could confirm the general trends that had been found while using genome-wide studies of circRNA abundance, namely that they are enriched in the brain when compared to other tissues [6,7], that they are more biologically stable than their linear counterparts [27], and that they become more abundant with age [6,33,34]. Moreover, we found that most of these circRNAs are also induced by ionizing radiation, although the timescale was different as compared to the linear transcripts. While mRNA expression quickly reduced following exposure, several circRNAs continued to rise even at later time points.

The delayed/persistent induction of these circRNAs after radiation exposure in comparison to that of their linear counterparts suggests that they are not involved in the early cellular response to radiation. In this respect, it is interesting to note that, in contrast to the circular isoforms of *Pvt1*, *Ano3,* and *Sec14l5*, which have no reported function in the DDR, the circRNAs originating from the DDR gene *Rnf169* show little, if any, induction after irradiation. This agrees with the general assumption that circRNAs are mostly involved in slow, long-term processes, such as cellular differentiation [35], rather than immediate-early responses [36], or perhaps the DDR itself.

This delayed and prolonged effect after irradiation may also have important practical implications. The early transcriptional response to radiation is a process that is almost entirely regulated by the activation status and level of P53, which in its turn depends on residual DNA-damage. This renders this response inherently transient and most of the P53-dependent changes in gene expression have therefore returned to the baseline levels within 24 to 48 h after exposure [37,38]. At the same time, these early changes in gene expression after irradiation, mostly from blood cells, have been shown to be very practical for the purpose of retrospective biodosimetry, in which radiation doses are predicted based on the gene expression profiles [3,39]. However, one of the main drawbacks of gene expression for biodosimetry is the transient nature of the effect [37]. We observed long-term induction of circular transcripts of Pvt1 in the blood of whole-body irradiated mice, while the mRNA was unchanged. In contrast, circular and linear Sec14l5 and linear, but not circular, Rnf169 transcript levels were reduced. Although the reduced expression of Sec14l5 and Rnf169 mRNA as such may also be useful for biodosimetry, most if not all of the genes that have been used for effective dose reconstruction are upregulated after radiation exposure [40]. With respect to *Pvt1*, it is furthermore interesting to note that this has been previously used for biodosimetry in mice [38,41] and ex vivo human blood [42,43], and may therefore represent a genuine radiation biomarker. Therefore, our observation of a prolonged radiation-induced response of circRNA levels may offer an important advantage over traditional mRNA profiling. However, this should be confirmed in studies that cover an expanded panel of radiation doses and time points in samples that are more suitable for biodosimetry, such as blood or lymphocytes. The potential suitability of circRNAs as a novel RNA species for biodosimetry would also strengthen previous findings that radiation-induced expression of splice variants of certain radiation-responsive genes offer enhanced sensitivity for biodosimetry [43].

Whether circRNAs may play a role in the late response to radiation-induced DNA-damage is currently unknown. One study showed that the irradiation of HUVEC cells (endothelial cells) changes the abundance of hundreds of circRNAs, of which the majority are transcribed from host genes that are involved in P53 regulation [44]. Furthermore, this study demonstrated that, after irradiation of U2OS *QKI-5* expression is induced, which thereby affects its association to the KIRKOS-71 and KIRKOS-73 circRNAs that subsequently showed increased expression and enrichment in exosomes via which they could potentially communicate with other cells [44]. This was not found in SHEP neuroblastoma cells [44], which may explain why the knockdown of Qki in Neuro-2a cells did not result in altered Pvt1 circRNA expression in this study, and that suggest cell type-dependent regulation of circRNA biogenesis.

PVT1 is a lnc-RNA that is expressed from the 8q24 “gene desert” locus adjacent to the *MYC* oncogene with which it interacts. It has gained much attention in recent years after it was demonstrated to contribute to ovarian and breast cancer pathogenesis independently from *MYC* [45]. Termed “a rising star among oncogenic lnc-RNAs” [46], it is now considered as a potential oncogene in different cancer types [23], in part by inducing cancer cell invasiveness [31,32]. Much of its effects depend on its ability to act as a competing endogenous RNA via the binding of microRNAs [47]. Additionally, circular transcript variants of *PVT1* have been shown to act as microRNA sponges in various cancers [48,49]. Whether the mouse ortholog *Pvt1* functions in a similar way is currently unknown.

Furthermore, our results corroborate previous findings that circRNAs are generally enriched in the brain as compared to other tissues, and that they are dynamically expressed during brain development and in differentiating neurons [6,7]. The continued increase in abundance of circRNAs with increased age (up to 220-fold for circ-0009172) as compared to the more unchanged expression of mRNA suggests that these circRNAs become the most abundant RNA species produced from these genes in the adult brain. As such, their accumulation may represent a biomarker of brain age [34], although it is currently unclear whether this also has functional consequences [33].

## Figures and Tables

**Figure 1 cells-08-00778-f001:**
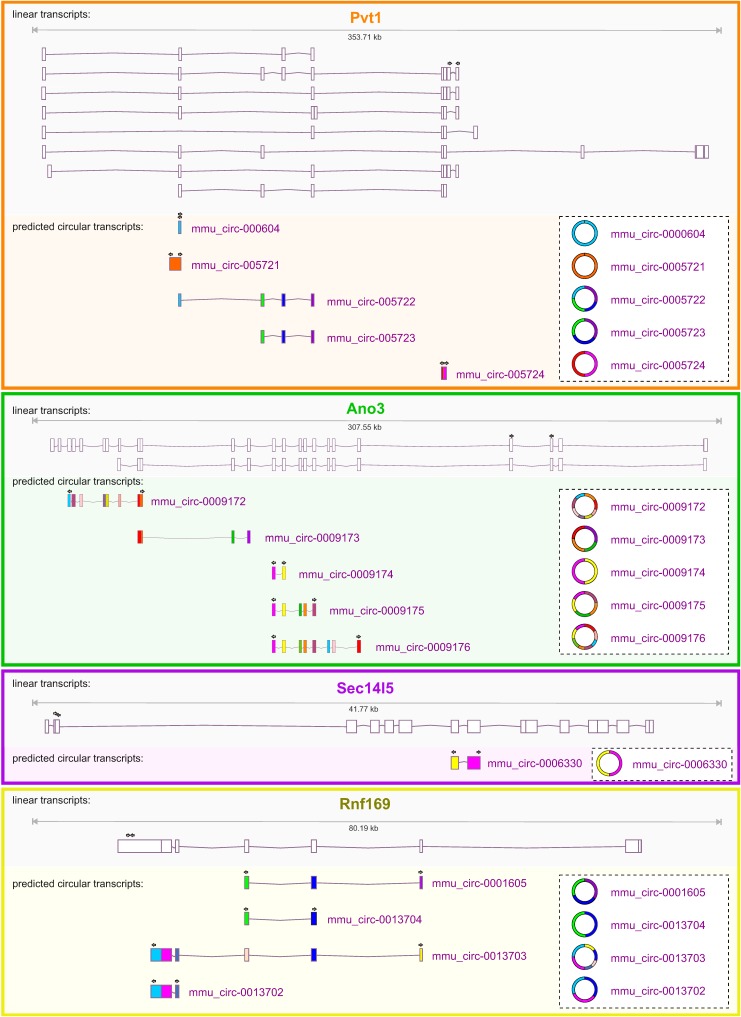
Predicted circRNAs transcribed from mouse *Pvt1*, *Ano3*, *Sec14l5,* and *Rnf169*. UCSC genome browser views. For each gene, the upper tracks represent mRNA splice variants. The lower tracts represent the predicted circRNAs based on Memczak et al., 2013 and Rybak-Wolf et al., 2015 [6,16]. Primers used for qRT-PCR are indicated by arrows. Please note that *Ano3* and *Rnf169* are transcribed from the-strand and are therefore represented in the 3’ to 5’ orientation.

**Figure 2 cells-08-00778-f002:**
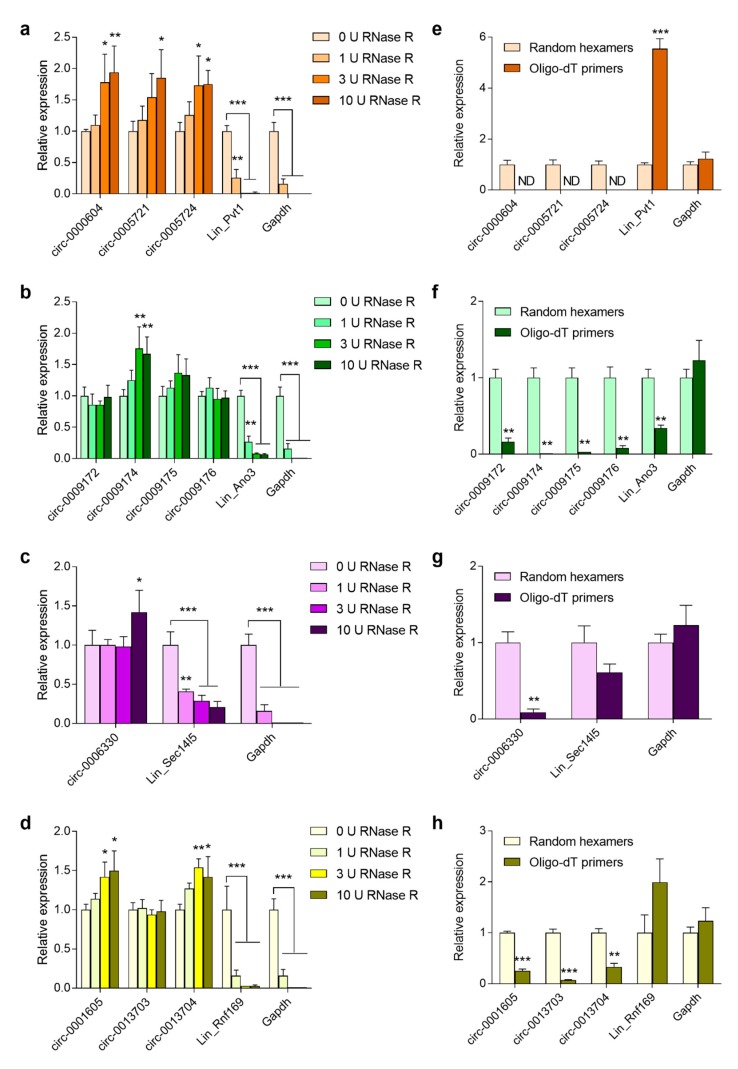
Experimental validation of predicted circular RNA candidates. (**a**–**d**) Expression of linear and circular transcript variants of *Pvt1*, *Ano3*, *Sec14l5,* and *Rnf169* was assessed via RT-qPCR using divergent primers and was performed on RNaseR treated RNA samples from mouse brain at P10; (**e**–**h**) The RNA was reverse transcribed using random hexamer vs. oligo-dT primers. Circular transcripts are compared to their respective linear counterparts and the linear mRNA of Gapdh. Data shown are means + s.e.m. of 4 biological replicates (*n* = 4) and were normalized against (**a**–**d**) 0 U RNaseR or (**e**–**h**) random hexamers. Statistics were performed using Student’s t-test with Holm–Sidak correction for multiple comparisons * *p* < 0.05; ** *p* < 0.01; *** *p* < 0.001. ND: Not detected.

**Figure 3 cells-08-00778-f003:**
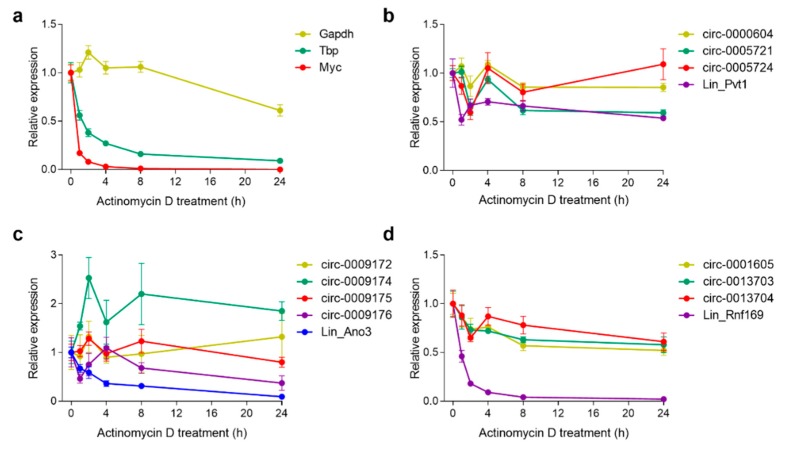
RNA stability at different times points after Actinomycin-D treatment. Relative expression levels of (**a**) control genes (*Tbp*, *Myc* and *Gapdh*); (**b**) *Pvt1*; (**c**) *Ano3*; and, (**d**) *Rnf169* transcript variants in primary cortical neurons at 0, 1, 2, 4, 8, and 24 h of treatment with Actinomycin-D. Data shown are means ± s.e.m. of 4 biological replicates (*n* = 4).

**Figure 4 cells-08-00778-f004:**
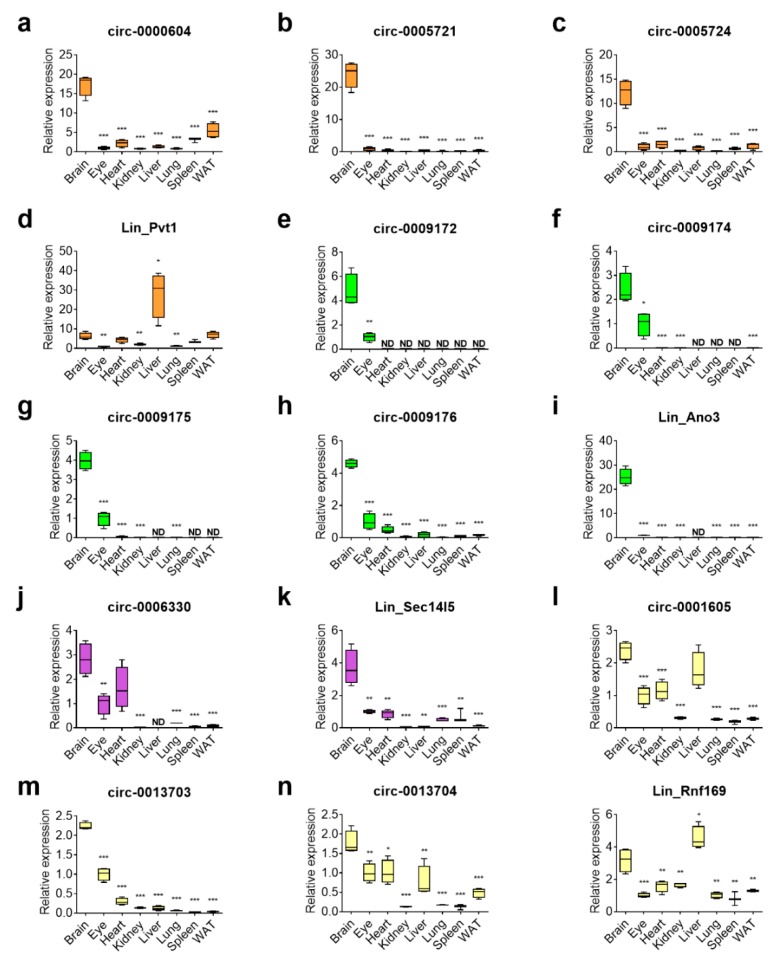
Profiling of circRNA expression across mouse tissues reveals enrichment in the brain. RT-qPCR analysis was performed to investigate the tissue distribution of (**a**–**d**) *Pvt1*; (**e**–**i**) *Ano3*; (**j,k**) *Sec14l5;* and, (**l**–**o**) *Rnf169*. The expression levels for each transcript variant are given as box plots for eight indicated tissues and normalized against expression in the eye (set at 1.0). Centerlines show the median; boxes represent the range between first and third quartiles and whiskers represent the highest and lowest values of four biological replicates (*n* = 4). Note that in some tissues the expression levels were very low compared to the brain, but still above background. For comparisons between the brain and other tissues, a one-way ANOVA with Dunnett’s correction for multiple comparisons was used; * *p* < 0.05; ** *p* < 0.01; *** *p* < 0.001. WAT: white adipose tissue; ND: not detected.

**Figure 5 cells-08-00778-f005:**
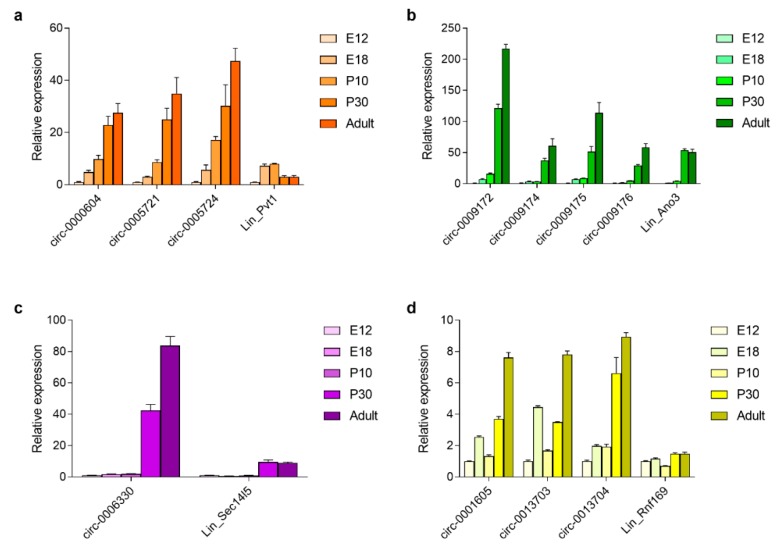
RNA expression during embryonic and post-natal brain development. Relative expression of linear and circular transcripts of (**a**) Pvt1; (**b**); Ano3; (**c**) Sec14l5; and, (**d**) Rnf169 in the brain at different developmental stages. Expression at E12 was set at 1. Data shown are means + s.e.m. of four biological replicates (*n* = 4).

**Figure 6 cells-08-00778-f006:**
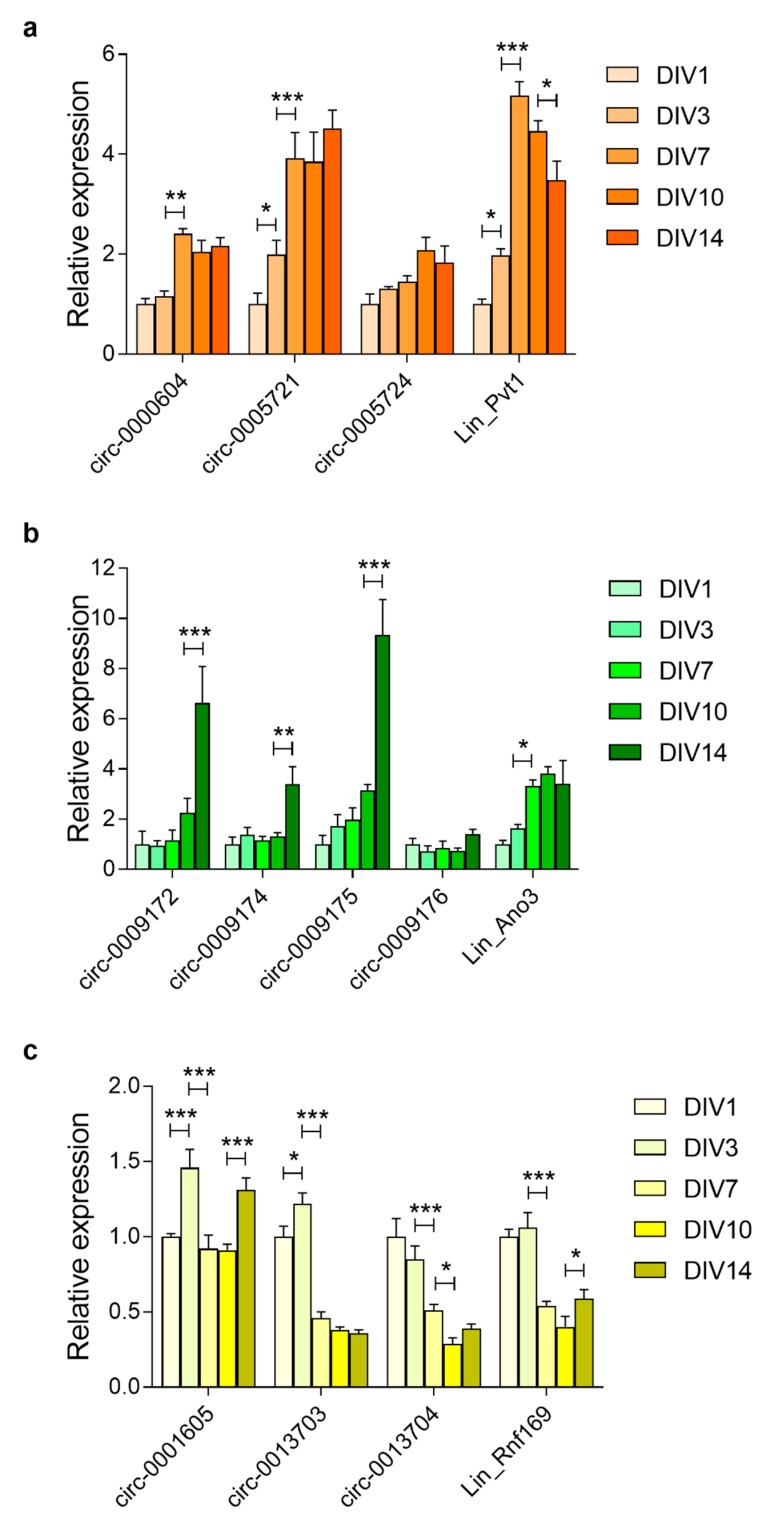
RNA expression during maturation of primary cortical neurons. Relative levels of linear and circular transcripts of (**a**) Pvt1; (**b**); Ano3; (**c**) Sec14l5; and, (**d**) Rnf169 in the primary cortical neurons at different days in vitro (DIV). Expression at DIV1 was set at 1. Data shown are means + s.e.m. of four biological replicates (*n* = 4). Statistics were performed using two-way ANOVA; * *p* < 0.05; ** *p* < 0.01; *** *p* < 0.001.

**Figure 7 cells-08-00778-f007:**
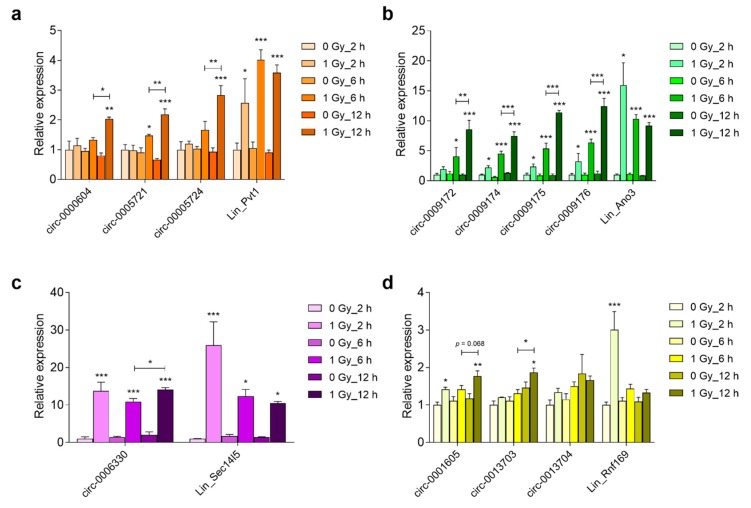
Radiation-induced RNA abundance in the embryonic brain. Relative levels of linear and circular (**a**) *Pvt1;* (**b**)*; Ano3;* (**c**) *Sec14l5,* and (**d**) *Rnf169* was determined by RT-qPCR in the brain 2 h, 6 h, and 12 h after irradiation of E11 embryos. Data shown are means + s.e.m. of three biological replicates and have been normalized against 0 Gy_2 h. Statistics were performed using two-way ANOVA with correction for multiple comparisons while using the two-stage step-up method of Benjamini, Krieger, and Yekutieli; * *p* < 0.05; ** *p* < 0.01; *** *p* < 0.001.

**Figure 8 cells-08-00778-f008:**
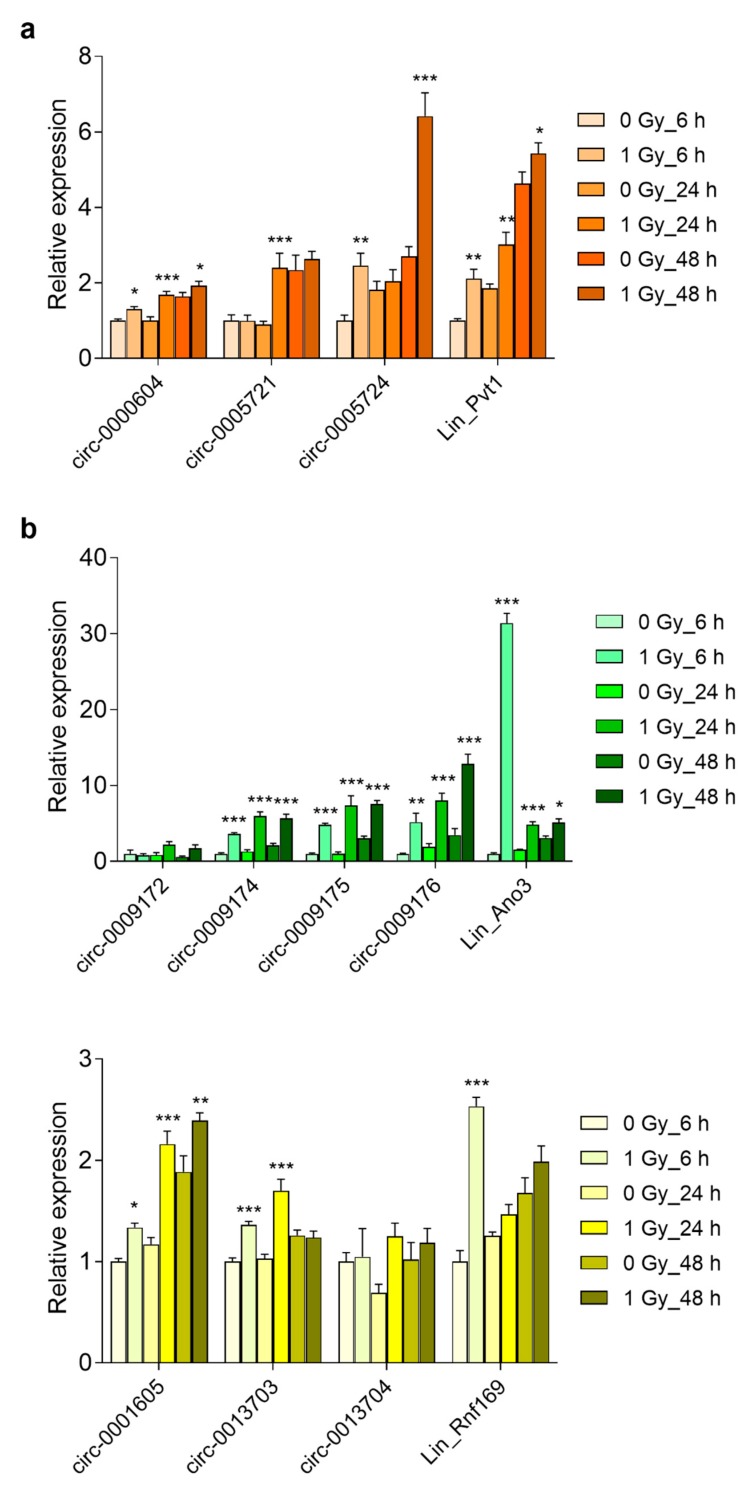
Radiation-induced RNA abundance in primary cortical neurons. Relative levels of linear and circular (**a**) Pvt1; (**b**); Ano3; (**c**) Sec14l5; and, (**d**) Rnf169 were determined by RT-qPCR at 6 h, 24 h, and 48 h after irradiation of DIV1 primary cortical neurons. Data shown are means + s.e.m. of three biological replicates (*n* = 3) and have been normalized against 0 Gy_6 h. Statistics were performed while using two-way ANOVA with correction for multiple comparisons using the two-stage step-up method of Benjamini, Krieger, and Yekutieli; * *p* < 0.05; ** *p* < 0.01; *** *p* < 0.001.

**Figure 9 cells-08-00778-f009:**
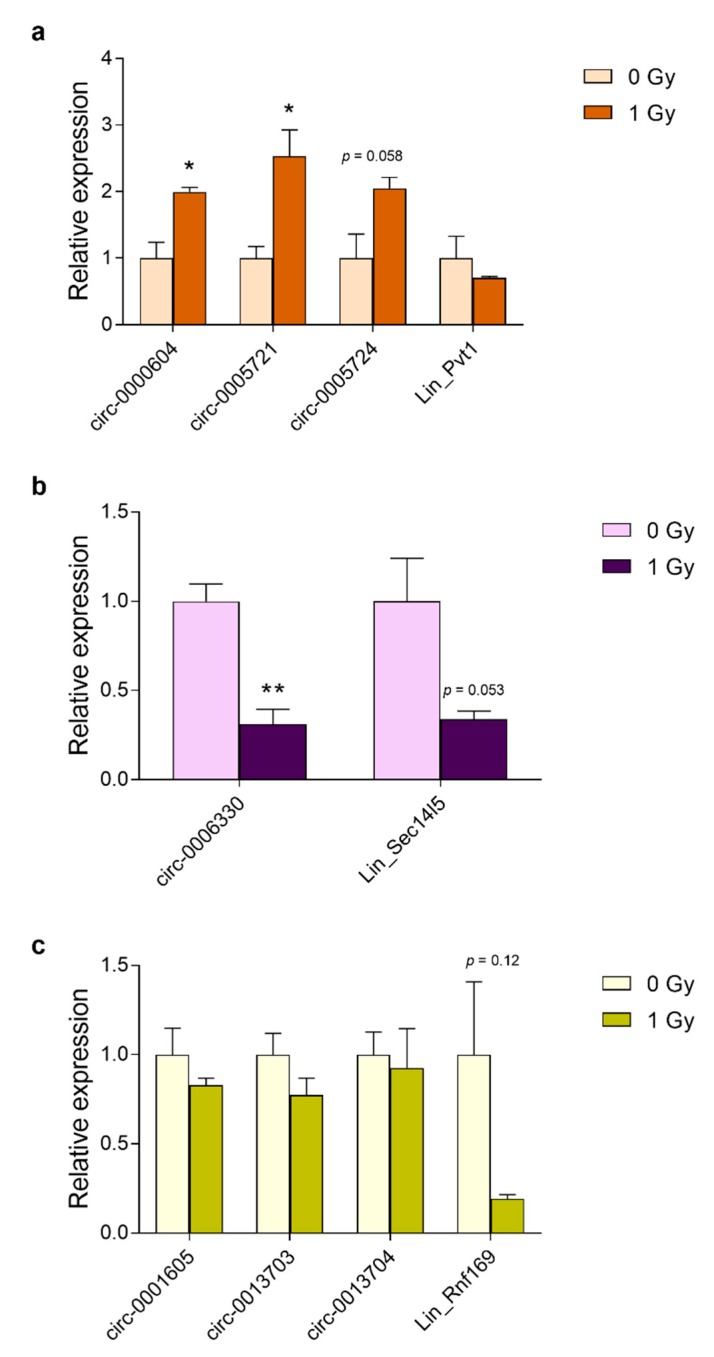
Radiation-induced RNA abundance in mouse blood. Relative levels of linear and circular (**a**) Pvt1; (**b**); Sec14l5; and, (**c**) Rnf169 transcripts were determined by RT-qPCR at 10 days after whole body irradiation of female mice. Data shown are means + s.e.m. of three biological replicates (*n* = 3) and have been normalized against 0 Gy. Statistics were performed using Student’s *t*-test; * *p* < 0.05; ** *p* < 0.01.

**Figure 10 cells-08-00778-f010:**
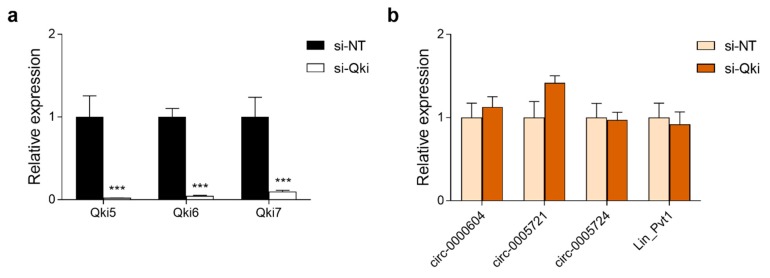
Silencing of the circRNA biogenesis regulator Quaking does not affect expression of *Pvt1* circRNAs in Neuro2a cells. (**a**) Qki isoforms were effectively silenced by Qki siRNA (si-Qki) in comparison to non-targeting siRNA (si-NT). (**b**) No effect of Qki silencing on expression of Pvt1 transcripts. Data are means + s.e.m. of 4 biological replicates (*n* = 4). Statistics were performed using Student’s *t*-test; *** *p* < 0.001.

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
