# Peer review of "Exposure to Ionizing Radiation Triggers Prolonged Changes in Circular RNA Abundance in the Embryonic Mouse Brain and Primary Neurons"

_cells, 2019, doi:10.3390/cells8080778_

Round 1
Reviewer 1 Report
A brief summary (one short paragraph) outlining the aim of the paper and its main contributions.
The manuscript describes a set of interesting and new data on the effects of exposure to ionising radiation on the pattern of expression of circular RNAs in the embryonic brain.
Broad comments highlighting areas of strength and weakness.
Everything is more or less fine, but the authors should clarify one important issue. It does not fallow from the text what stats was used in this study and whether the authors have corrected all the p-values by multiple testing (Bonferroni correction?). This should be clearly stated.
Author Response
The manuscript describes a set of interesting and new data on the effects of exposure to ionising radiation on the pattern of expression of circular RNAs in the embryonic brain.
Broad comments highlighting areas of strength and weakness.
Everything is more or less fine, but the authors should clarify one important issue. It does not fallow from the text what stats was used in this study and whether the authors have corrected all the p-values by multiple testing (Bonferroni correction?). This should be clearly stated.
We thank the reviewer for pointing this out. Indeed, we did not mention for every analysis which multiple testing correction had been used. We have now adapted this in the revised manuscript. Statistical tests are always mentioned in the figure legends.
Reviewer 2 Report
This manuscript describes a study that examined the time course of expression of a total of 15 circular RNAs (circRNA), products of 4 P53 target genes, in mouse brain after in utero exposure to a low dose of X-rays (1 Gy) and in mouse neurons in vitro. At baseline, mouse brain had relatively high levels of cirRNAs compared to other tissues. Moreover, radiation caused a prolonged upregulation of several of the circRNAs, suggesting that circRNAs may be useful in biodosimetry to estimate the dose of ionizing radiation that an individual has previously been exposed. The manuscript is clearly written, and the finding of potential new biomarkers of radiation exposure could have an impact on the field of biodosimetry. A few comments remain.
Specific comments:
1. Abstract, line 21: Do the authors mean after radiation? If so, please add to the text for clarity.2. Introduction, line 45: please add the words “to radiation”.
3. Materials and methods:
a. Lines 92-93: Please indicate whether male or female post-natal and adult mice were used. Also, please discuss the potential of the influence of sex on radiation-induced circRNA induction.
b. Line 96: The word “in” is used twice.
c. Line 100: replaced “by” the plating medium ...
d. Lines 101 and 109: Please make sure that the final article will have superscript numbers in 3x105 cells and 106 cells.
e. Line 107: For clarity, please describe that Neuro2a cells are of a mouse neuroblastoma cell line.
f. X-irradiation: Please include the field size of the X-ray beam. Please describe if mice were anesthetized during radiation and how. If not, please describe the holder that was used to restrain them in the X-ray beam.
4. Figures 2, 3, 5, 6, 7, 8, 9: The figure legends describe “replicates”. Do the authors mean the number of samples per group? Or the number of repeated reactions per sample? Please clarify.
5. Figure 4: Please include the number of mice per group in the figure legend.
6. Figure 5: Why were statistical tests not performed on the data in this figure?
7. Results, line 274: identified on “the” basis of ...
Author Response
This manuscript describes a study that examined the time course of expression of a total of 15 circular RNAs (circRNA), products of 4 P53 target genes, in mouse brain after in utero exposure to a low dose of X-rays (1 Gy) and in mouse neurons in vitro. At baseline, mouse brain had relatively high levels of cirRNAs compared to other tissues. Moreover, radiation caused a prolonged upregulation of several of the circRNAs, suggesting that circRNAs may be useful in biodosimetry to estimate the dose of ionizing radiation that an individual has previously been exposed. The manuscript is clearly written, and the finding of potential new biomarkers of radiation exposure could have an impact on the field of biodosimetry. A few comments remain.
We thank the reviewer for critically reading our manuscript and for the very useful comments and suggestions.
Specific comments:
1. Abstract, line 21: Do the authors mean after radiation? If so, please add to the text for clarity.
No, this was not after irradiation.
2. Introduction, line 45: please add the words “to radiation”.
Thank you for this remark. We have adapted the sentence.
3. Materials and methods:
a. Lines 92-93: Please indicate whether male or female post-natal and adult mice were used. Also, please discuss the potential of the influence of sex on radiation-induced circRNA induction.
We thank the reviewer for this valuable remark. However, all our initial experiments were performed on male mice because the gender effect was not a subject for investigation. For the additional experiment on mouse blood we have used female mice. These informations have now been added to the materials and methods section of the revised manuscript.
Thus, based on our current results we cannot make any meaningful speculation about the influence of gender on the radiation-induced expression of circRNAs.
b. Line 96: The word “in” is used twice.
Thank you for this remark. We have adapted the sentence.
c. Line 100: replaced “by” the plating medium ...
Thank you for this remark. We have adapted the sentence.
d. Lines 101 and 109: Please make sure that the final article will have superscript numbers in 3x105 cells and 106 cells.
Thank you for this remark. We have adapted the sentence.
e. Line 107: For clarity, please describe that Neuro2a cells are of a mouse neuroblastoma cell line.
Thank you for this remark. We have adapted the sentence.
f. X-irradiation: Please include the field size of the X-ray beam. Please describe if mice were anesthetized during radiation and how. If not, please describe the holder that was used to restrain them in the X-ray beam.
Thank you for this remark. We have now included the requested information in the Materials and Methods section of the revised manuscript. Field size is 31 cm diameter and mice were individually placed in a custom-made plexiglass irradiation pie.
4. Figures 2, 3, 5, 6, 7, 8, 9: The figure legends describe “replicates”. Do the authors mean the number of samples per group? Or the number of repeated reactions per sample? Please clarify.
We apologize for the lack of clarity. These are biological replicates. We have now clarified this in the figure legends of the revised manuscript.
5. Figure 4: Please include the number of mice per group in the figure legend.
We have now included this information in the revised manuscript.
6. Figure 5: Why were statistical tests not performed on the data in this figure?
We thank the reviewer for this remark. In fact, we did perform the statistical tests (using two-way ANOVA). However, there has been a lot of debate in recent years about the usefulness and validity of p-values and we felt that the results in this figure were sufficiently self-explanatory while adding asterisks would only add complexity to the figure because of the large number of comparisons.
7. Results, line 274: identified on “the” basis of ...
Thank you for this remark. We have adapted the sentence.
Reviewer 3 Report
The paper by Mfossa et al. describes a series RT-qPCR assays measuring the concentrations of transcripts of four mouse genes, with focus on their circularized forms, dependence on developmental stage, tissue of origin, time in culture, and irradiation. In most cases in the time-course experiments a progressive increase in circRNA abundance is observed, from which the authors conclude that expression increases with age and after a single exposure to ionizing radiation.
The observational nature of the study and the focus on four genes, makes it difficult for me to appreciate its general significance. Since all mechanistic inferences are speculative, and since there already appear to be much more extensive studies covering thousands of circRNAs by RNAseq followed by qPCR validation, this work may provide insights into the biology of the four genes studied, but in my view doesn’t go further than that. E.g., see the following papers about effect of irradiation (not cited by the authors)
PeerJ. 2018 Jun 15;6:e5011. doi: 10.7717/peerj.5011. eCollection 2018.
Comprehensive circular RNA expression profile in radiation-treated HeLa cells and analysis of radioresistance-related circRNAs.
Yu D1, Li Y1, Ming Z2, Wang H1, Dong Z3, Qiu L1, Wang T1.
Cell Physiol Biochem. 2018;47(6):2558-2568. doi: 10.1159/000491652. Epub 2018 Jul 10. Identification of Circular RNAs Altered in Mouse Jejuna After Radiation.
Lu Q, Gong W, Wang J, Ji K, Wang Y, Xu C, Liu Y, He N, Du L, Liu Q.
Leuk Res. 2018 Jul;70:67-73. doi: 10.1016/j.leukres.2018.05.010. Epub 2018 May 28.
Circular RNA expression profiles are significantly altered in mice bone marrow stromal cells after total body irradiation. Yu D1, Li Y1, Ming Z2, Wang H1, Dong Z3, Qiu L1, Wang T1.
These are from the top of list of matches, and only for one type of treatment.
The idea of using circRNA transcripts produced form P53-controlled genes for retrospective biodosimetry is very interesting, but this aspect of the paper would need to be further developed to make it convincing, since dose-dependence was not studied and only brain tissue was analysed (perhaps not the easiest one to get samples from for dosimetry). Also, age-dependence of circRNA abundance shown in the paper and potential variability between individuals make it unclear how practical this idea is.
Since the four genes were selected based on their p53 dependence, testing circRNA accumulation is dependent on p53 signalling would be a way to add mechanistic detail and make the study more meaningful, e.g. by using p53 knock-out mice. Use of DNA repair deficient lines could also help.
Methodology:
The main reason I cannot assess the methodological soundness of the results is the lack of proper definition of replicates. Does “4 replicates” mean four true biological replicates (independent RNA isolations from four separate animals) or something different (e.g. four qPCR experiments using the same cDNA prep)? The numbers of animals used in experiments is not stated.
Use of housekeeping genes has to be described better than “Gapdh and/or Polr2a served as internal reference genes” for each experiment. Since housekeeping gene expression can also vary, reliance on a single one, especially when comparing tissues, may be risky.
Data is presented as “relative expression”, however what serves as control is not always stated, and in some cases I have to guess what it might be, for example in the case of tissue profiling in Figure 4.
Other remarks:
I suggest to avoid the use of the term “circRNA expression”to describe the observations presented in the paper. Strictly speaking “expression” applies to genes. The term “RNA expression” is widely used colloquially, and in other situations pointing this out would be merely pedantic, but in the case of hyper-stable RNA species, I think it causes real confusion. It reads as if the transcriptional activity of the gene is gradually increasing with age or time in culture / after irradiation. However, if no substantial degradation of circRNA is happening (and this might be the case based on Figure 3), the concentration of circRNA could progressively increase without any change to the transcription or splicing. Since only the abundance of circRNA was measured, and not its production (≈expression) or degradation, the appropriate term should be used.
Actinomycin D experiments make me question the reliability of RNA abundance measurement. The time points are described as 0-24h “after treatment with actinomycin D”. Since the duration of treatment is not stated, my interpretation based on the purpose of the experiment is that the treatment was continuous for the indicated periods of time. If this is correct: Figure 3C circ-0009174 RNA is increasing concentration after transcription block — is this an evidence of the assay artifacts or can circRNAs be generated in the absence of transcription? The fluctuations in the curves (e.g. drops at 1–2h in panel D, followed by increases) also require explanation.
It would be useful to indicate primer locations on the scheme in Figure 1. Also in the PDF I received the image is of poor quality.
